# An Energy Data-Driven Approach for Operating Status Recognition of Machine Tools Based on Deep Learning

**DOI:** 10.3390/s22176628

**Published:** 2022-09-01

**Authors:** Wei Yan, Chenxun Lu, Ying Liu, Xumei Zhang, Hua Zhang

**Affiliations:** 1School of Automobile and Traffic Engineering, Wuhan University of Science and Technology, Wuhan 430081, China; 2Department of Mechanical Engineering, School of Engineering, Cardiff University, Cardiff CF24 3AA, UK; 3Hubei Key Laboratory of Mechanical Transmission and Manufacturing Engineering, Wuhan University of Science and Technology, Wuhan 430081, China; 4Academy of Green Manufacturing Engineering, Wuhan University of Science and Technology, Wuhan 430081, China

**Keywords:** operating status recognition, energy data-driven, deep learning, machine tools, fault diagnosis

## Abstract

Machine tools, as an indispensable equipment in the manufacturing industry, are widely used in industrial production. The harsh and complex working environment can easily cause the failure of machine tools during operation, and there is an urgent requirement to improve the fault diagnosis ability of machine tools. Through the identification of the operating state (OS) of the machine tools, defining the time point of machine tool failure and the working energy-consuming unit can be assessed. In this way, the fault diagnosis time of the machine tool is shortened and the fault diagnosis ability is improved. Aiming at the problems of low recognition accuracy, slow convergence speed and weak generalization ability of traditional OS recognition methods, a deep learning method based on data-driven machine tool OS recognition is proposed. Various power data (such as signals or images) of CNC machine tools can be used to recognize the OS of the machine tool, followed by an intuitive judgement regarding whether the energy-consuming units included in the OS are faulty. First, the power data are collected, and the data are preprocessed by noise reduction and cropping using the data preprocessing method of wavelet transform (WT). Then, an AlexNet Convolutional Neural Network (ACNN) is built to identify the OS of the machine tool. In addition, a parameter adaptive adjustment mechanism of the ACNN is studied to improve identification performance. Finally, a case study is presented to verify the effectiveness of the proposed approach. To illustrate the superiority of this method, the approach was compared with traditional classification methods, and the results reveal the superiority in the recognition accuracy and computing speed of this AI technology. Moreover, the technique uses power data as a dataset, and also demonstrates good progress in portability and anti-interference.

## 1. Introduction

Intelligent computer numerically controlled (CNC) machine tools are necessary pieces of equipment for the furthering the development of modern productivity [1]. According to incomplete statistics, the number of machine tools in China has exceeded 8 million [2]. A CNC machine tool is a complex system integrating mechanical, electrical, hydraulic and other technologies. At the same time, some studies have shown that the complexity of the system structure leads to the cascading of its failures. Even if the failure rate of one component is low, the overall failure rate of the system increases exponentially due to the scale effect of complex systems. This renders maintenance of machine tools increasingly difficult. The fault diagnosis of machine tools in operation plays an irreplaceable role in ensuring the running quality of machine tools, especially in maintaining the level of machining accuracy [3]. Therefore, improving the fault diagnosis capability will significantly improve the life cycle of the machine tools, and has become a central focus within relevant industries and academia. When a machine tool fails, it is necessary to identify the time of failure and the failed energy-consuming unit as soon as possible. Fu et al. [4] found that this time accounted for 80–90% of the total fault diagnosis time, which was not conducive to subsequent fault diagnosis.

The OS of the machine tools is influenced by their energy-consuming units. By accurately recognizing the OS of the machine tool, it is possible to accurately analyze the energy-consuming units that are working at any time. This reduces the time required for locking the time point and location of the fault, which plays an important role in improving the efficiency of system fault diagnosis and represents preparatory work for fault diagnosis. Existing research shows that the operating state (OS) of a machine tool can be divided into cutting, starting, standby, etc. [5,6]. This paper uses the power curve of the XK713 milling machine operating process, as shown in Figure 1. The energy-consuming units contained within the power of each OS can be visually observed. For example, in the cutting state, almost all the energy units of a machine tool are running, such as the spindle drive system, feed drive system, lubrication and cooling system, etc. and only three energy units are in a standby state. When the machine tool fails, by determining the OS of the machine tool, the energy-consuming unit where the failure is located can be identified. However, due to the instability of the machining site and the accuracy of the measuring instruments, it is difficult to establish a specific model to accurately recognize the OS of the machine tool. For example, the load factor is a key factor in calculating the energy consumption in the cutting state, which needs to be obtained by fitting the experimental data of additional sensors, which is difficult to apply in actual power production [7]. To this end, a deep learning method with power data is proposed to recognize the OS of machine tools in this paper, which solves the problems of large amounts of machine tool data and a complex working environment, and demonstrates good recognition accuracy and generalization ability.

The rest of the paper is structured as follows. Section 2 describes the related work for the OS recognition of a machine tool. Section 3 introduces the methodology of the proposed approach, based on the general framework establishment, whereby the three main steps, namely power data acquisition, OS recognition with deep learning technology ACNN and model validation are also studied. In Section 4, a case study is applied to verify the feasibility and superiority of the proposed method. Then, conclusions and further research directions are outlined in Section 5.

## 2. Literature Review

An essential basis for this study is to classify and recognize the OS of a machine tool. In general, the OS is a term to describe the series of activities of machining tools [8,9], and can be divided into five stages, namely Starting status (S1), Standby status (S2), Idling of the spindle status (S3), Air cutting status (S4) and Cutting status (S5) [10]. With a defined boundary of OS, several researchers focus on the energy-consumption nature of machine tools and their energy-consuming units. Oliver et al. divided the OS of a machine tool into the steady state and transient state, and estimated the energy consumption of the spindle and feed axis with these states [11,12]. Lv et al. established a spindle acceleration state model of CNC lathes based on a calculation of the moment of inertia for the spindle drive system [13]. Martin et al. conducted an experimental study to investigate the energy characteristics required for obtaining the power models of CNC machine tools through an experimental study [14]. These studies provide the fundamental theory for this work.

When the boundaries and energy-consumption characteristics of OS are determined, the power thresholds can be set to recognize the OS of a machine tool. There are two main approaches for power threshold acquisition in the existing studies. One is to measure the key parameters of the OS energy consumption, and the other concerns calculation of the power thresholds in each OS by using the parameters [15]. Hu et al. designed a spindle speed experiment to obtain the non-load power factor, which was used to calculate the power thresholds of S4 [16]. As a key parameter for S5, the load-loss coefficients during the machine tool service process (MTSP) were calculated by using a data-fitting approach, and the data were collected from the torque sensor, power sensor and power meters in the machining experiment [17]. Benjamin et al. proposed a Gaussian threshold method to adjust the judgment threshold for different machine tool OS recognition [18]. Liu et al. designed a machining experiment of the machine tool spindle drive system, and calculated the load-loss coefficients [19]. The aforementioned power-threshold approaches provide strong support for understanding the energy-consumption characteristics of OS and the complex relationship with energy-consuming units, which contribute to the study of OS recognition in this paper. Unfortunately, most of the threshold methods require installation of additional sensors in the processing system, but are difficult to widely implement in actual industrial production processes and also lack generalizability.

The other is to establish the multi-period energy-consumption model of energy units, where the power thresholds of OS could then be obtained by summing the results of these models. Due to the complex structure of machine tools, the energy-consumption models of the feed drive system [20,21], spindle drive system [22,23] and fluidic systems [24] were established. He et al. combined the colored timed object-oriented Petri net (CTOPN) and the virtual component method to establish the dynamic characteristic model of multiple energy sources of CNC machine tools [25]. Jia et al. established a kinetic energy-consumption model [26]. The above two models are based on the energy consumption of the basic actions of machine tools, and the energy consumption of the OS can be obtained by the addition of the two values of consumption. The above approaches often depend on a large number of experiments to improve the recognition accuracy, and are inconvenient for implementation and operation in actual production. Furthermore, the number of energy units is increasing due to the continuous structural improvements of machine tools, and the above approaches are often labor-intensive and computationally costly.

Traditional machine tool fault diagnosis mainly uses signal processing and machine learning techniques. The signal processing techniques used in machine tool fault diagnosis mainly include time-domain analysis [27], frequency-domain analysis [28] and time-frequency analysis. Wavelet Analysis [29], short-time Fourier transform (STFT) [30] and empirical mode decomposition are commonly used methods in the time-frequency analysis of machine tool signals. In recent years, the convolutional neural network (CNN) has been used in the field of image recognition. Great achievements have been achieved in this domain, which can be extracted from the original complex and robust multi-dimensional features that are extracted from raw data [31,32], so that the established model has higher accuracy and robustness. Wan et al. [33] proposed an improved two-dimensional LeNet-5 network for monitoring and identification of rolling bearing faults. Shi et al. [34] use the DCNN algorithm to identify the effective operating state of the machine tool. In the above methods and models, the adaptive feature extraction of original data by CNN only considers the multi-dimensional characteristics of the data, and does not consider the problems of data generalization and gradient explosion, which result in losses of original data feature information. Rohit et al. [35] use a modified AlexNet in conjunction with the Grasshopper Optimization Algorithm (IGOA) to diagnose faults present in gearboxes and to correct the faulty gear. In addition, after the classification process, performance evaluation is performed on various performance indicators. Existing deep learning [36] methods are processed and optimized to improve its performance. However, due to the constraints of experimental conditions, the evaluation cost is higher, and large deep convolutional networks such as VGGNet and ResNet-18 that use more memory and parameters (140 M) have not been adopted [37].

Therefore, this paper proposes a data-processing method based on WT, which only needs to collect power data during operation of the machine tool, and the collection method is convenient and fast. Further, the proposed method does not require the use of additional hardware measurements. It can be used as a general method to determine the current OS of various machine tools, thus ensuring its practical application in manufacturing enterprises. Finally, the paper adopts an ACNN model with more hidden layers, which employs multiple GPU operations [38] and an advanced activation function ReLU, capable of extracting deeper features from the data.

## 3. Materials and Methods

### 3.1. General Framework

The general framework of the proposed approach includes three main stages, power data collection and processing, recognition model establishment and model validation, as shown in Figure 2.

Power data collection and processing. The purpose of this stage is to obtain the input of ACNN, which mainly includes three parts: power data acquisition, power data preprocessing and feature extraction.Recognition model establishment. In this stage, an AI recognition model for OS of machine tools is established with ACNN. Then, the adjustment approach for hyperparameters of ACNN is also studied to improve the recognition performance.Model validation. To illustrate the feasibility of the above model, a confusion matrix approach is employed to verify the recognition accuracy.

### 3.2. Power Data Collection and Processing

#### 3.2.1. Power Data Collection and Preprocessing

The dataset in this study comes from the Green Manufacturing Laboratory of Wuhan University of Science and Technology. We use the power tester to collect the power data of the entire processing process of the milling machine XK713 from start to finish, that is, the three-phase current signal during processing, and then display it on the host computer. In order to avoid the influence of a single number of processing machines on the test and to increase the stability of the acquisition environment, the entire test process was carried out on 20 machine tools of the same model. A total of 400 blank milling tests were performed.

In addition, it should be noted that there are various sources of noise present in the original current signal, which may cause serious interference to the main components of the current signal and reduce the quality of the data set. In order to improve the signal-to-noise ratio, this paper uses a rectangular window function to smooth the original data. Considering the power curve of the OS of machine tools, the sine function could be used as the time-domain prototype formula of the filter window, as shown in Equation (1).
(1)$$y=\frac{\sin(2\pi fx)}{\pi x},$$
where, f is the cut-off frequency of filter, which is used to describe a special frequency of the frequency characteristic index; x is the current signal value collected; y is the signal value after filtering and transformation.

In addition, due to the time-series characteristics of the current signal, the time-domain formula of the filter window is constructed with Equation (2).
(2)$$X(k)\Rightarrow x(i)\cdot \frac{\sin(2\pi f(i-\frac{L-1}{2}))}{\pi (i-\frac{L-1}{2})},$$
where, L represents the window function width; i represents the sample value sequence number, and i∈[0,L−1]; X(k) is the filter sequence; x(i) is the current sampling point. L and x(i) depend on the step size and sampling frequency and are the key to smooth denoising of the raw data.

#### 3.2.2. Feature Extraction

The correct input form of the ACNN is a prerequisite to ensure the reasonable use of the entire model. Time-domain characteristics can reflect the timely and dynamic processing of the machine tool and the trend and characteristics of signal changes during processing. Therefore, the continuous WT approach is employed to extract features of the processed signal to time-frequency images, and these images can be used as input for the ACNN. The continuous WT function is shown in Equation (3).
(3)$$\psi (\alpha ,\tau )=\frac{1}{\sqrt{\alpha }}\int_{-\infty }^{\infty }f(t)\cdot \psi (\frac{t-\tau }{\alpha })\cdot dt,$$
where, ψ(α,τ) is the wavelet function; α is the scale factor; τ is the translation factor and f(t) represents the commonly used piecewise function, and is the collected current signal in this paper.

Subsequently, the wavelet time-frequency images are stored in .jpg format, and resized to the pixel-based dimensions of 227 × 227, which can be used as input for the ACNN.

#### 3.2.3. Data Augmentation

The time-frequency diagram of the processed current signal is intercepted. Next, 5 different categories of sample images are generated (corresponding to 5 OS, respectively), each group of 400 images, a total of 2000 images.

In order to improve the adaptability and generalization of the network model, this study uses random rotation, horizontal and vertical flipping, adjusting brightness, saturation and contrast and adding Gaussian blur to the obtained images to expand the dataset [39]. The number of augmented datasets is 8 times that of the original, with a total of 16,000 pictures. Examples of time-frequency pictures under different expansion methods are shown in Figure 3. In order to ensure a balanced sample size, 2000 images of different categories were randomly selected from each type of samples, totaling 10,000 images. According to the ratio of 8:2, they are divided into a training set (8000 pictures) and test set (2000 pictures).

### 3.3. Recognition Model Establishment

#### 3.3.1. Network Construction

The AlexNet Convolutional Neural Network (ACNN) is an improved algorithm of the traditional Convolutional Neural Network, which uses training techniques such as the ReLU activation function, overlapping pooling, dual GPU training, data augmentation and dropout, and satisfies the requirements of power data processing [40]. The ACNN was firstly proposed in the Large-scale Visual Recognition Challenge (ILSVRC 2012) [41], which is a deep convolutional neural network based on the classic LeNet5 and the traditional BP neural network [42], with the network construction shown in Figure 3.

Now, we are ready to describe the overall architecture of our ACNN. As shown in Figure 4, the convolutional layer has a total of five layers (C1–C5), and the fully connected layer has a total of three layers (FC1–FC3).

The input data of the first layer (C1) is the original 227 × 227 × 3 image, which is convolved by a 11 × 11 × 3 convolution kernel, and the moving step size is 4 pixels. (This is the receptive field of adjacent neurons in the kernel map [43] distance between centers). The second convolutional layer (C2) takes as input the (response normalization and pooling) output of the first convolutional layer and filters it with 256 kernels of size 5 × 5 × 48. The input data of the second layer is the pixel layer output by the first layer. In order to facilitate subsequent processing, the left and right sides and the upper and lower sides of each pixel layer should be filled with 2 pixels [44]; Operations are then performed on different GPUs. Each set of pixel data is convolved with 5 × 5 × 48 convolution kernels, and a total of 256 kernels of size 5 × 5 × 48 are used to filter them. The third, fourth and fifth convolutional layers (C4, C5) are connected to each other without any intermediate pooling or normalization layers. The third convolutional layer has 384 kernels of size 3 × 3 × 256 connected to the (normalized, pooled) outputs of the second convolutional layer. The fourth convolutional layer has 384 kernels of size 3 ×3 × 192, and the fifth convolutional layer has 256 kernels of size 3 × 3 × 192. Fully connected layers have 4096 neurons each.

The last three layers are the fully connected layer (FC), which is employed to expand the multi-dimensional matrix data generated by the convolution of the last convolution layer into one dimension, and predict each category.

(1)Convolutional layer

The main goal of the convolutional layer is to extract features from the input image. On the convolutional layer, multiple convolution kernels are used to convolve the input image, and after adding paranoia, a series of output feature maps can be obtained through a nonlinear activation function. This layer has characteristics of local connection and weight sharing, both of which reduce the complexity of the model and reduce the number of parameters. Among them, taking the convolution of a single GPU of the C1 convolution layer as an example, the convolution process of AlexNet is shown in Figure 5 and Equation (4).
(4)$$X_{j}^{l}=f(\sum_{i\in M_{j}}X_{i}^{l-1}\cdot w_{ij}^{l}+b_{j}^{l}),$$
where, $X_{j}^{l}$ is the element of the second layer; $M_{j}$ is the fourth convolution area of the *j*-layer feature map; $X_{i}^{l-1}$ is the element in it; $w_{ij}^{l}$ is the weight matrix corresponding to the convolution kernel; $b_{j}^{l}$ is the paranoid term. *f* is the activation function ReLU.

(2)Pooling layer

The main function of the pooling layer is to reduce the dimension of the feature map while maintaining the invariance of the feature scale to a certain extent. Pooling layers reduce the number of parameters and computation by gradually reducing the size of the representation space to control overfitting. Pooling layers usually take a convolutional layer as input. The pooling process is shown in Figure 6 and Equation (5).
(5)$$X_{j}^{l}=f(\beta_{j}^{l}down(X_{i}^{l-1})+b_{j}^{l}),$$
where, *down*() is the subsampling function; $\beta_{j}^{l}$ is the weight of the *j*-feature map of the *l*-layer; $b_{j}^{l}$ is the bias of the *j*-feature map of the *l*-layer.

(3)Fully Connected layer

After the input image is alternately propagated through multiple convolutional layers and pooling layers, the fully connected layer network is used to classify the extracted features. In the fully connected layer, the one-dimensional feature vectors expanded by all feature maps at the input are obtained by weighted summation and passing through the activation function. The calculation process is shown in Equation (6).
(6)$$y^{k}=f(w^{k}x^{k-1}+b^{k}),$$
where, *k* is the serial number of the network layer; *y^k^* is the output of the fully connected layer; *x^k^*^−1^ is the expanded one-dimensional feature vector; *w^b^* is the weight coefficient; *f* is the activation function.

(4)Regularization

Due to the high complexity of deep learning models, large-scale training data are critical for model robustness. However, in the tool wear monitoring problem, it is difficult to obtain large-scale training data. Therefore, this paper introduces ReLU [45], LRN [46], and Dropout [47] into the ACNN model training to regularize the network.

Nonlinear activation function ReLU. Its gradient descent is always 1, which can effectively prevent the problem of gradient disappearance and shorten the training time. This operation can be represented by Equation (7).


(7)
f(x)=max(0,x),


Local Response Normalization. Local response normalization (LRN) creates a competition mechanism for the activity of local neurons, rendering the response values increasingly large, thus inhibiting other neurons with smaller responses, enhancing the generalizability of the model.Dropout. Randomly deleting some neurons through a defined probability, while keeping the number of neurons in the input layer and output layer unchanged, and update the parameters. Repeat these operations in the next iteration until the end of training. Dropout can effectively prevent overfitting of neural networks.

#### 3.3.2. Network Training and Parameter Adjustment

The parameters and regression layer feature weights in the ACNN model need to be learned through model training, and thus the model needs to be trained through training data to obtain the optimal parameters in the model. The difference between the predicted value and the true value for a particular sample is defined as loss. Here, the authors choose to use an Adaptive moment estimation (Adam) [48] to minimize the SoftMax categorical cross-entropy loss function, the loss function is defined as Equation (8).
(8)$$Loss=-\frac{1}{N}\sum_{i=1}^{N}{y^{\prime }}_{i1}lny_{i1}+{y^{\prime }}_{i2}lny_{i2}+\ldots +{y^{\prime }}_{iM}lny_{iM},$$
where, *N* represents the number of samples; *M* represents the number of classes; *y*′ represents the predicted output value; *y* displays the actual value.

Model predictions where the model has been tested against a given test dataset by collecting power signals. During testing, there are some hyperparameters for stochastic gradient descent for training convolutional neural networks. In the proposed ACNN model, two main hyperparameters are investigated to improve the recognition accuracy.

The initial learning rate [49]. The initial learning rate controls how well we adjust the network weights and error convergence based on the gradient of the loss.Batch size [50]. The stochastic gradient descent method is used in training convolutional neural network, and its batch size affects computer memory utilization and training oscillation which represents the number of training examples in a single batch. The batch size limits the number of training samples before each weight update.

### 3.4. Model Validation

The ACNN is essentially a deep learning-based classifier, the classified performance of which could be verified with the data classification validation models [51]. In this paper, the confusion matrix is used to evaluate the classified performance of the ACNN.

For the classification task M, the number of samples is N(N=2000), $C=\{C_{1},C_{2},\ldots ,C_{L}\}$ is the classifier set (i.e., ACNN), and the L(L=1) classifiers are used to test in the samples N respectively. The confusion matrix $C_{k}(k=1,2,\ldots ,L)$ of each classifier is obtained, and the confusion matrix of the kth classifier $C_{k}$ is shown in Equation (9):(9)$$C_{k}=\left[\begin{array}{cccc} N_{1,1}^{k} & N_{1,2}^{k} & \cdots & N_{1,M}^{k}\\ N_{2,1}^{k} & N_{2,2}^{k} & \cdots & N_{2,M}^{k}\\ \vdots & \vdots & N_{i,j}^{k} & \vdots \\ N_{M,1}^{k} & N_{M,2}^{k} & \cdots & N_{M,M}^{k} \end{array}\right],$$
where, the elements in the i-th row and the j-th column represent the number of the i-th class identified as the j-th class by the classifier $C_{k}$ in the sample. If i=j, it means that the classifier can correctly recognize the number of samples, so the diagonal elements represent the classifier $C_{k}$.

The number of correct classifications, and the off-diagonal elements represent the number of misclassifications by the classifier $C_{k}$. If, ideally, both the recall and precision of the classifier are 100%, then the off-diagonal elements of the confusion matrix are all 0, and only the diagonal elements are non-zero.

In this study, the accuracy rate (Accuracy, %) was used to evaluate the performance of all different OS classification models, and the misclassification of five OS was analyzed by confusion matrix. The accuracy is the ratio of the number of correctly classified samples to the total number of samples, which is calculated as Equation (10)
(10)$$Accuracy=\frac{TP+TN}{TP+TN+FP+FN}\times 100\%,$$
where, *TP*, *FP*, *FN*, and *TN* are the statistics of the classification of different OS by the classification model in the confusion matrix, respectively. Among them, *TP* is the number of samples judged as positive in the positive class, *FP* is the number of samples judged as positive in the negative class, *FN* is the number of samples in the positive class judged as the negative class, and *TN* is the number of samples in the negative class judged as the negative class. When performing classification tasks, the number of real categories of samples to be predicted is regarded as the number of positive samples, and the sum of all other categories is the number of negative samples.

## 4. Case Study

### 4.1. Experimental Conditions

In this paper, a face milling experiment with the milling machine XK713 is designed to verify the proposed model and approach. In this case, the WT1800 power tester is used to measure three-phase current signal during machining, and the different machining parameters (cutting speed n, feed amount $f_{v}$ and back cut amount $a_{p}$) are also set up to illustrate the generalization ability of the proposed approach.

Test environment for ACNN model training: hardware includes Intel(R) Core(TM) i7-6500U CPU@2.5GHz processor with 20 GB of memory and an NVIDIA GeForce RTX 940M graphics card. Software includes operating system Ubantu22.04 (64-bit), programming language Python3.9, Deep learning framework Pytorch1.12.0, general computing architecture CUDA10.2 and GPU acceleration library CUDNN8.5.0. The main processing equipment and parameters are shown in Figure 7 and Table 1, respectively.

### 4.2. Results and Discussion

#### 4.2.1. Results

With the proposed WT approach, the time-frequency analysis results for different OS current signals of the XK713 are obtained, and the time-frequency images for each state are shown in Figure 8a–e, respectively. It can be seen that there are obvious differences in the time-frequency images of the current signals for the five OS, and the time-frequency images can be used to recognize the OS of XK713.

To study the tuning method of two hyperparameters, namely initial learning rate and batch size, the Adam optimizer was used to set different initial learning rates and batch sizes to train the model, and the effects of different parameters on the accuracy of the model were compared and analyzed. The variation trend of the accuracy of the test set with the parameters is shown in Figure 9a,b.

Considering the influence of the experimental environment, blindly pursuing high batch size and low initial learning rate cannot achieve the expected effect. When the learning rate is less than 0.0005 or greater than 0.002, the loss value has an upward trend. When the learning rate is set within 0.001, the ACNN network has the lowest loss value with high accuracy. When the batch size value is greater than 64 or less than 8, the accuracy rate has a downward trend; when the batch size is set to 16, ACNN has the highest accuracy rate and the lowest loss value. Therefore, the optimized values for the initial learning rate and batch size are 0.001 and 16.

After selecting the optimal hyperparameters, the classification performance of the model is evaluated using the confusion matrix, and the results are shown in Figure 10.

In the confusion matrix, the rows and columns of the confusion matrix represent the True label and Output label of each condition [52]. Figure 10 shows the confusion matrix of the test sample used in the improved ACNN, giving detailed classification results.

As shown in Figure 7, the overall recognition accuracy of S3 is the highest, and the highest error rate is only 1% (only 1% of S5 is recognized as S3). The reason may be that the current fluctuates violently when the machine tool is turned on, and its characteristics are easier to identify. However, the overall recognition accuracy of S4 is relatively low, indicating that ACNN misclassifies S4 the most. Even 2.6% of S4 were misidentified as S2, the highest among all experiments. It is foreseeable that the current fluctuations of S4 and S2 are not clear, the gap between the peak and the trough is not large, and can easily be erroneously classified. In addition, S1 has the most balanced recognition accuracy, with the highest recognition error rate (S3) and the lowest recognition error rate (S4) differing by 0.8%. In general, the recognition accuracy of the five states can reach about 97%, the highest can reach 97.8%, the lowest can reach 96.5%, and the peak difference is only 1.3%. It can be seen that the method combining WT and ACNN not only has high classification accuracy, but also has relatively stable classification performance.

Furthermore, by further analyzing the misclassification of different machine tool operating system datasets, this study shows that the color and texture features of different OS are important basis in the classification process. Misclassification is likely to occur when there is noise. The misclassified samples are mostly images with high brightness, low saturation and added Gaussian blur. Its complex background interferes with the characteristics of the image to some extent, which affects the accurate identification of the sample.

#### 4.2.2. Discussion

To further illustrate the superiority of the proposed model, this paper uses the following four common models to analyze from the perspective of recognition accuracy and computational cost: Linear Regression (LR), Backpropagation (BP), LeNet-5 and Residual Network (ResNet-18). The first three models bear fewer network layers and are more commonly used early classification models, and the fourth model is currently a more popular classification model. To investigate the performance of the proposed feature extraction method, three typical feature extraction methods were selected for comparison: engineering methods, statistical methods, and principal component analysis (PCA). For computational load, there are two methods of computational load that can be used to analyze ACNN models:

$T_{1}$: Time spent on modeling during training;$T_{2}$: The time taken to generate the sample.

Each model was run 20 times, using the same initial learning rate, batch size, and training set, and the final average was taken as the result. Table 2 shows the corresponding results for the five classifiers. It can be seen that the maximum classification accuracy rates are 85.17%, 89.17%, 91.56%, 97.89% and 98.14%, respectively.

As shown in Table 2, compared with the linear model, the excellent performance of the deep learning model in model recognition shows that the feature extraction of the original data by the deep learning model can mine deeper and more comprehensive feature information, which proves the feasibility and effectiveness of Deep Learning Models. With the increase in the network structure, the accuracy of ACNN is much higher than that of LeNet-5, and it is not much inferior to ResNet-18. On the contrary, when calculating the load, the time spent by ResNet-18 is multiplied compared to that of ACNN, and the impact on the server and the actual production process is self-evident.

It can also be seen from the table that LR requires the least computational time to develop the model in the milling machine. In contrast, the proposed ACNN is computationally expensive because ACNN spends a lot of time on convolution and pooling operations. ResNet-18 obviously increases the computational cost significantly after adopting the residual structure. Average computation time for generating samples. The longest one is ResNet-18, because a lot of time is spent on updating features. Unexpectedly, ACNN is almost the same as BP and LeNet-5 when adopting the PCA feature lifting method

By comparison, it can be found that the fitting effect of the ACNN model is significantly better than that of other classification models. The main advantages of proposed model are:The training parameters are adjusted. Because the adjusted model network has a suitable batch size, it can aggregate faster. In particular, the complex classification of the energy consumption of machine tools is especially notable. The data collection time is short, and easily affected by the processing environment;Feature extraction techniques such as wavelet decomposition are combined with deep learning networks. The performance went well on small training samples, the network is very sensitive to the characteristics of the data, and it is not prone to loss and overfitting;The structure of the model. The ACNN model facilitates real-time data processing without human intervention. Moreover, the operation speed is improved by using ReLU and multiple CPUs. At the same time, techniques such as overlapping pooling, data gain, and dropout are used to improve the operational accuracy of ACNN and reduce overfitting.

## 5. Conclusions

In this paper, an ACNN-based current data-driven identification method of a machine tool operating system is proposed. The identification of the operating system is a precondition for the fault diagnosis of the machine tool, and is of great significance to the fault diagnosis of the machine tool. This study only requires raw current data to identify the machine tool operating system and mine sensitive energy consumption features from the data. Compared with the traditional threshold method and experimental method, the generalization is stronger, and the requirements for the scene environment and artificial feature extraction are not high. In addition, this paper extracts the temporal features of the data through WT, which reduces the noise in the process of converting the data to grayscale images. While meeting the input requirements of ACNN, more accurate feature information can be included. Finally, a series of experiments were carried out. The results show that compared with linear regression, BP, LeNet-5 and RESNET-18, classifiers, ACNN performs better than the above-mentioned classifiers in most cases. Furthermore, it has faster recognition speed and higher recognition accuracy.

The proposed ACNN provides an efficient, simple and fast method to determine the operating system of a machine tool. The method has certain guiding significance for reducing maintenance time, improving system reliability and ensuring the safe operation of machine tool systems. In the real-world production environment, the collected machine tool current data are becoming increasingly complex. At present, the model has only been verified in milling processing, and its application in turning and other processing can be considered in the follow-up research. This can be aided by increasing the scale of model training data to further improve the accuracy of the mode, improving the test environment, and adopting more advanced AI algorithms to improve the recognition performance.

## Figures and Tables

**Figure 1 sensors-22-06628-f001:**
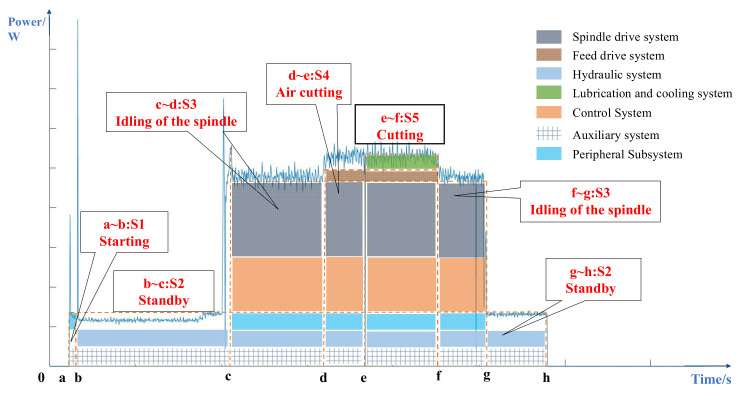
Power curve of the XK713 machine tool process.

**Figure 2 sensors-22-06628-f002:**
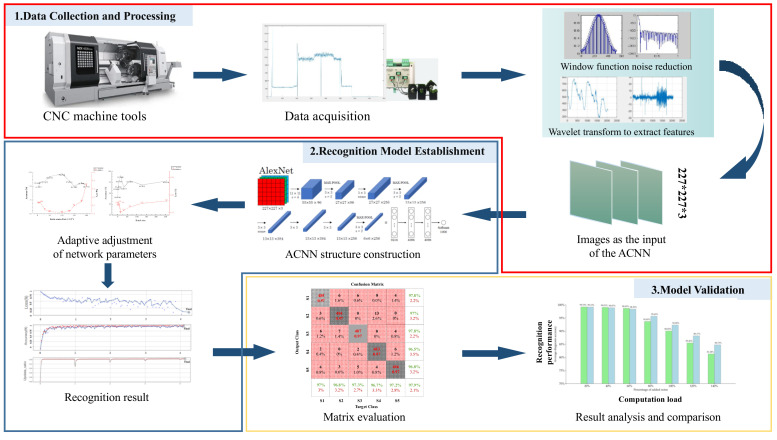
The general framework of the proposed approach.

**Figure 3 sensors-22-06628-f003:**
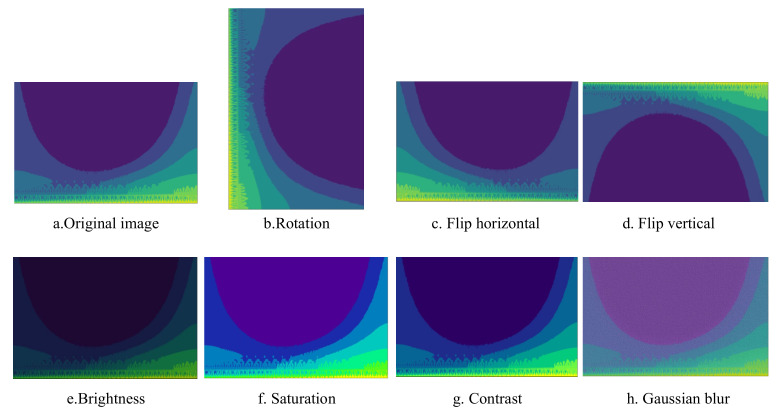
Example of machine tool OS images after data augmentation.

**Figure 4 sensors-22-06628-f004:**
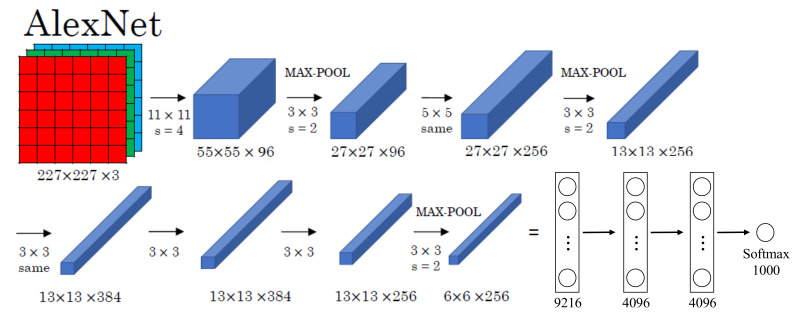
Overall framework for the improved network.

**Figure 5 sensors-22-06628-f005:**
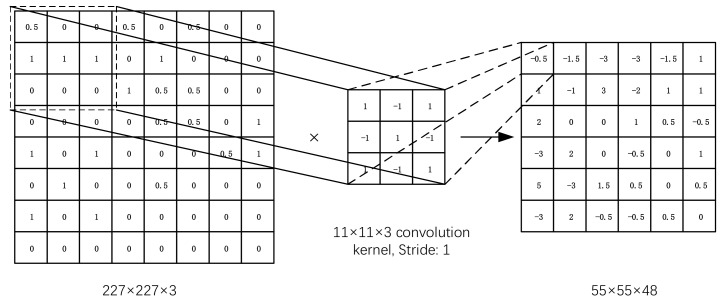
The process of Convolution.

**Figure 6 sensors-22-06628-f006:**
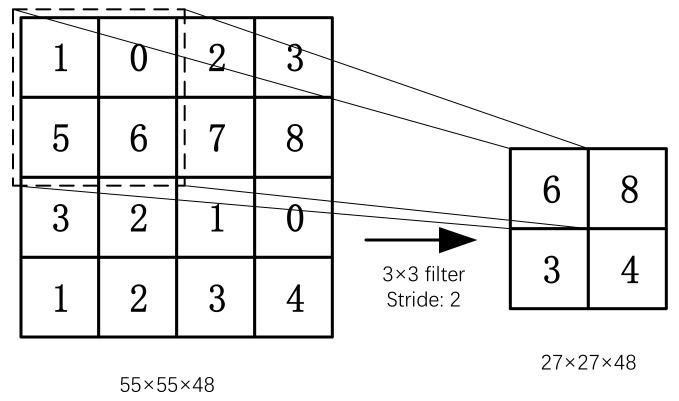
The process of Pooling.

**Figure 7 sensors-22-06628-f007:**
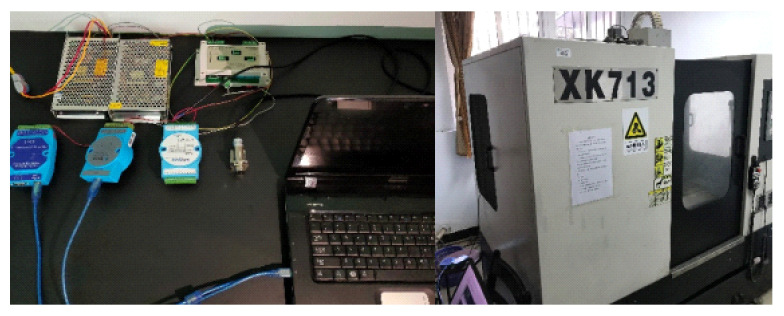
Equipment used at the experimental site.

**Figure 8 sensors-22-06628-f008:**
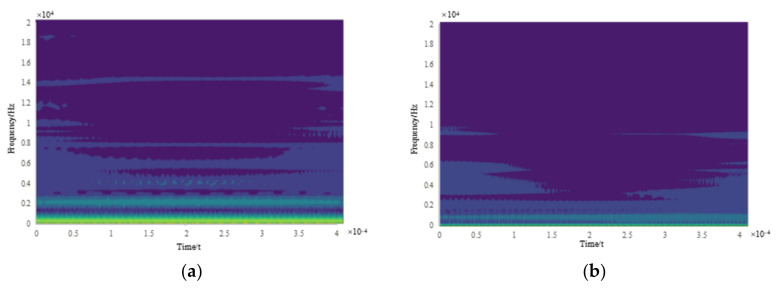
The variance curve of the detail signal D3 obtained by wavelet decomposition of the W-phase electric current signal. (**a**) S1 state wavelet decomposition time-frequency image; (**b**) S2 state wavelet decomposition time-frequency image; (**c**) S3 state wavelet decomposition time-frequency image; (**d**) S4 state wavelet decomposition time-frequency image; (**e**) S5 state wavelet decomposition time-frequency image.

**Figure 9 sensors-22-06628-f009:**
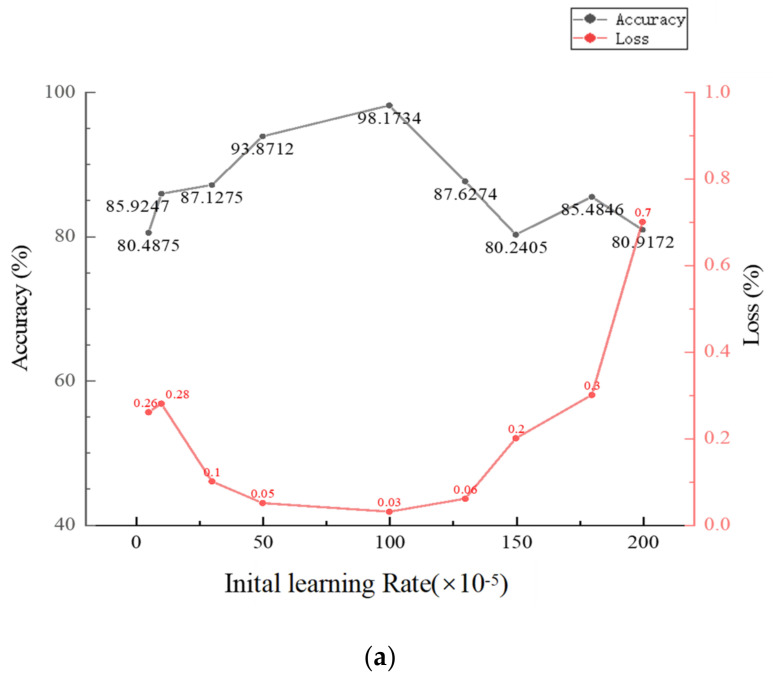
The effect of hyperparameters on accuracy and loss. (**a**) The effect of initial learning rate value on accuracy value and loss; (**b**) The effect of batch size value on accuracy and loss.

**Figure 10 sensors-22-06628-f010:**
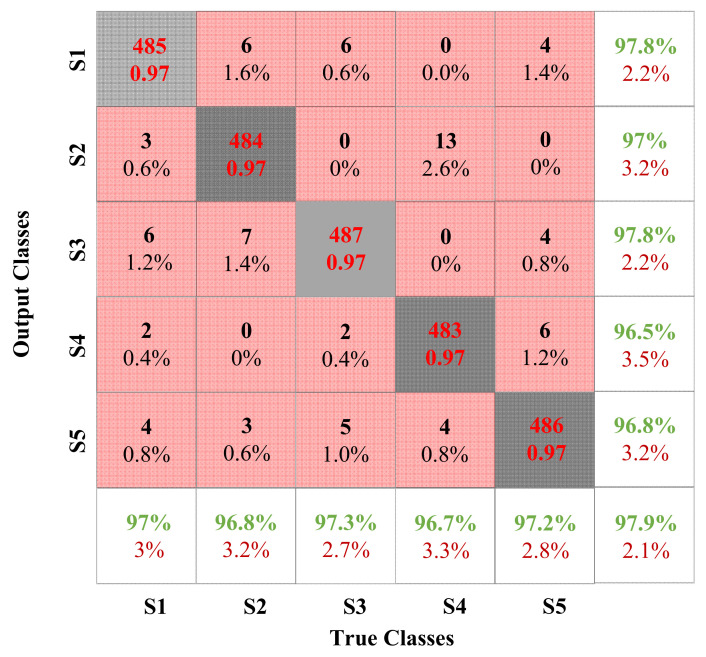
Confusion matrix of the proposed approach.

**Table 1 sensors-22-06628-t001:** Main parameters of cutting activity.

Activities	Tool Diameter	n (r/min)	$f_{v}\;(\mathrm{mm}/a)$	$a_{p}\;(\mathrm{mm})$	$L_{cutting}\;(\mathrm{mm})$	$L_{air}\;(\mathrm{mm})$
1	12	800	225	1.5	600	200
2	12	450	150	6	464	32
3	12	450	150	2	600	45
4	12	950	262.5	0.2	582	56
5	12	950	262.5	6	324	28
6	12	600	187.5	2	159	72
7	12	600	187.5	2	159	72

**Table 2 sensors-22-06628-t002:** Accuracy of different models and time spent modeling and generating samples.

Type of Comparison		Linear Regression	BP	LeNet-5	ACNN	ResNet-18
**Training times**		20	20	20	20	20
**Initial learning rate**		/	/	0.01	0.01	0.01
**Accuracy (%)**		85.17	89.17	91.56	97.89	98.14
**T_1_**	**EST**	0.016	0.055	0.028	0.040	0.184
	**STAT**	20.357	28.387	30.489	39.224	111.874
	**PCA**	16.872	17.5780	20.784	33.874	109.477
**T_2_**	**EST**	0.014	0.041	0.046	0.102	0.544
	**STAT**	0.014	0.058	0.027	0.093	0.578
	**PCA**	0.026	0.034	0.029	0.031	0.129

## Data Availability

The datasets used and/or analyzed during the current study are available from the corresponding author on reasonable request.

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
