# Peer review of "An Energy Data-Driven Approach for Operating Status Recognition of Machine Tools Based on Deep Learning"

_sensors, 2022, doi:10.3390/s22176628_

Round 1
Reviewer 1 Report
This paper proposes a new method of energy data-driven for operating status recognition of machine tools based on deep learning. An effective model, the AlexNet Convolutional Neural Network, is introduced in this paper. By using the time-frequency images obtained by the continuous wavelet transform of three-phase current signal from machine as the inputs of the network, the model can accurately identify operating status recognition of machine tools, which provides a good idea for the time series classification problems based on CNN. However, there are also some problems, which must be solved before it is considered for publication:
1) Relevant research background needs to be supplemented, it is necessary to add a review of existing deep learning-based methods for operating status recognition of machine tools.
2) There are some structural errors in the article, the Session 2 should have a detailed description of the related works, such as the mathematical principles of convolution, pooling, activation functions and normalization methods.
3) The innovation of the article is insufficient. The proposed ACNN model is a simple variant of the AlexNet. It is recommended to propose an innovative and effective model.
4) It is recommended to select the SOTA methods based on deep learning for operating status recognition of machine tools in recent years as the baseline for comparison, for highlighting the advantages of the model.
5) A more detailed description of the specific structural parameters of the ACNN model is required.
6) A more detailed description of the specific structure, hyperparameters, optimizer, loss function, and metrics of the ACNN model is required.
7) More detailed description of the test data set and preprocessing method is required.
8) The design of the experiment is not reasonable enough, such as the selection of batchsize.
9) Another obvious problem with this paper is lack of sufficient explanation of the simulation results. You need to explain your simulation results in detail and why you got such results.
10) Some sentences contain grammatical or format mistakes, such as, in page 3, “time domain analysis [27], Frequency Domain Analysis [28] and time-frequency analysis” would be “time domain analysis [27], frequency domain analysis [28] and time-frequency analysis”.
11) The work is about machine tools. Thus, the publication on machine tools should be cited. For example:
a)An iterative learning method for realizing accurate dynamic feedforward control of an industrial hybrid robot. Science China Technological Sciences, 2021, 64(6): 1177-1188.
b)An overview of dynamic parameter identification of robots. Robotics and Computer Integrated Manufacturing, 2010, 26(5): 414-419.
c)Mechatronics modeling and vibration analysis of a 2-DOF parallel manipulator in a 5-DOF hybrid machine tool. Mechanism and Machine Theory, 2018, 121, 430-445.
Reviewer 2 Report
The paper is well written and organized except of few comments for the improvement of the paper:
The abstract needs to be re-written in a coherent way to reflect the contributions of the presented research.
Few typo errors, please re-check.
Table 2 needs to be cited in the text
Section 4.2, there are few typo errors where the authors cited Table 4 instead of Table 2.
Title of Table 2 needs to be corrected.
The reference list needs to be updated.
Reviewer 3 Report
This work presents a technique for operating mode identification. The work is nicely presented. The overall progress of the paper is sound. The results give good reasoning behind the work too. I enjoyed reading it. The paper may be accepted for publication upon addressing the following comments.
1. The conclusion should clearly state the limitations of the work and some potential suggestions to improve it.
2. This is an international journal. The reference looks like the literature is congested to a specific geographic region only, which is not the actual picture. I would suggest authors amend their reference list.
3. The datasets should be made available through accessible public media for the promotion of the field and reproducibility.
Good luck!
Round 2
Reviewer 1 Report
The paper has been revised. However, there are some errors in Reference [7]. The reference page is wrong and the family name of the authors are not right.
